# Endophytic Microorganisms in Tomato Roots, Changes in the Structure and Function of the Community at Different Growing Stages

**DOI:** 10.3390/microorganisms12061251

**Published:** 2024-06-20

**Authors:** Yufei Wei, Siyu Chen, Xinyan Zhou, Diancao Ding, Jingjing Song, Shangdong Yang

**Affiliations:** 1Guangxi Key Laboratory of Agro-Environment and Agro-Products Safety, National Demonstration Center for Experimental Plant Science Education, Guangxi Agricultural College, Guangxi University, Nanning 530004, China; wyf18177217289@163.com (Y.W.); 2017391004@st.gxu.edu.cn (S.C.); zhouxinyan0923@163.com (X.Z.); 15177258605@163.com (D.D.); 2Guangxi Key Laboratory of Beibu Gulf Marine Biodiversity Conservation, College of Marine Sciences, Beibu Gulf University, Qinzhou 535011, China

**Keywords:** tomato (*Solanum lycopersicum*), endophytic, bacteria, fungi, growing stages

## Abstract

This study analyzed flower bud differentiation and fruiting stages to investigate how the structure of the plant endophytic microbial community in the roots of tomatoes changes with plant senescence. Based on high-throughput sequencing technology, the diversity and relative abundance of endophytic microorganisms (bacteria and fungi) in tomato stems at different growth stages were analyzed. At the same time, based on LEfSe analysis, the differences in endophytic microorganisms in tomato stems at different growth stages were studied. Based on PICRUSt2 function prediction and FUNGuild, we predicted the functions of endophytic bacterial and fungal communities in tomato stems at different growth stages to explore potential microbial functional traits. The results demonstrated that not only different unique bacterial genera but also unique fungal genera could be found colonizing tomato roots at different growth stages. In tomato seedlings, flower bud differentiation, and fruiting stages, the functions of colonizing endophytes in tomato roots could primarily contribute to the promotion of plant growth, stress resistance, and improvement in nutrient cycling, respectively. These results also suggest that different functional endophytes colonize tomato roots at different growth stages.

## 1. Introduction

Tomatoes are important vegetables, and improving their yield and quality has proved to be an urgent task in modern China [1]. As it has been established, extending the harvest period of vegetables is a good way to increase vegetable yields, for example, in asparagus and broccoli [2,3]. In other words, a higher tomato yield may be achieved with a longer harvest time and anti-aging effects [4].

The endophytic environment refers to specific locations within plant organs or tissues, where microorganisms exert neutral, detrimental, or beneficial effects on their host plants [5]. Endophytes have garnered attention because of their capacity to positively influence plant growth and their indirect role as biocontrol agents [6]. A succession of microbiomes throughout the growth cycle plays a pivotal role in the life history of plants. However, the dynamics of the endophytic microbial communities during succession are poorly understood [7]. Studies on the dynamics of rhizosphere microbial communities during plant development have revealed that plant transitions may be divided into the accumulation of random resources in the early stages and high-density functionally complementary communities in later stages, which could provide resistance against bacterial pathogens [8,9].

The necessary conditions for the survival, succession, and colonization of different microbial taxa can change with the development of plants, as observed in the rhizosphere and phyllosphere microbiomes, which vary with growth stage or age [10,11]. Core microbiomes, such as the root and leaf tissues of wheat [12], tomato leaves [13], leaves of Arabidopsis [14], and leaves and roots of *Boechera stricta* [15], could be altered. Robinson et al. reported that significant differences in endophytic bacteria could be detected in wheat during various growth stages [16]. The community structure of endophytic bacteria exhibits different nutrient distributions as plants age [6]. Plant tissues are additionally colonized by fungi [16], with an increase in the relative abundance of fungi such as *Chaetosphaeria* [17]. Moreover, the proliferation of fungi inhibits endophytic bacteria such as *Streptomycetaceae* [18] and *Burkholderiaceae* [19].

Age-related immunity is a senescence-associated immune response that is microbiome-dependent [14]. Previous studies have also confirmed that cytokinin-mediated immunity related to microbial contents varies with the age and developmental status of plants; cytokinin levels are significantly lower in mature plant tissues than in seedlings [20]. It has also been suggested that higher cytokinin levels in seedlings might support more abundant bacterial communities within the seedling stage. However, the elevated cytokinin levels decrease over time as the plants mature. In Arabidopsis, a potential decline in senescence-related bacteria can also be detected with decreasing cytokinin levels [21]. As cytokinin levels fundamentally alter senescence in plant development, it can support plant meristematic tissues, such as in rice and Arabidopsis [22], in delaying senescence and preserving ‘youthfulness’, viz., anti-aging, at high cytokinin levels.

Different dominant microbes present in tomato roots at various developmental stages and have beneficial effects, such as growth-promoting characteristics [8,23]. For instance, photosynthetic bacterial communities at different growth stages demonstrate significant differences, with the highest functional diversity of photosynthetic bacteria being observed at the seedling stage [24]. Fruit maturation is additionally regulated by multiple factors, including a metabolic network coordinated by environmental factors, physiological factors, and various signaling molecules and hormones [25]. Moreover, significant changes have been detected in the accumulation of carbohydrates, free amino acids, organic acids, and volatiles during maturation [26].

We hypothesized that the endophytic microorganisms in tomato roots at different developmental stages could assemble in different community structures. Therefore, this study aimed to explore how differences in the endophytic microbial compositions in tomato roots change at different developmental stages.

## 2. Materials and Methods

### 2.1. Experimental Site Description and Design

The experiment was conducted from September 2020 to March 2021 in the greenhouse of the experimental base of the College of Agriculture of Guangxi University (108°17′ E, 22°51′ N). The highest temperature in the site was 34 °C, the minimum temperature was 16 °C, and the average temperature was 22 °C. The tomato variety Zhongyan 868(R) was used in this study. The seeds were purchased from Zhongyan Yinong Seedling Technology Co., Ltd., Beijing, China. The experimental pots (height: 35 cm, diameter: 50 cm) were filled with 20 kg of soil. The soil type of the study area was acid red loam. Before planting, there were no plants on the soil surface.

A total of three groups were established to sow and raise seedlings on 10 September, 10 October, and 10 November 2021, with 15 plants planted in each group. The groups comprised the seedling stage (GY) (30 d after sowing), the flower bud differentiation stage (GH) (60 d after sowing and flowering), and the fruiting stage (GJ) (100 d after sowing and fruiting). The tomato root samples from each group were collected simultaneously on 20 December 2021, at the GY, GH, and GJ stages.

### 2.2. Soil Physicochemical Properties

Soil pH value was determined with a pH meter (soil/water ratio: 2/5, *w*/*v*); and the soil organic matter (SOM) content was determined by an external heating method using potassium dichromate. Soil total nitrogen (TN) content was quantified using the Kjeldahl acid digestion method; soil total phosphorus (TP) content was quantified using the molybdate blue method after acid digestion. Soil total potassium (TK) was determined using alkali fusion flame spectrophotometry; soil available nitrogen (AN), phosphorus (AP), and potassium (AK) were measured using the alkali diffusion method, double acid method, and flame photometry, respectively [27].

The pH was 5.7 and the contents of organic matter and total nitrogen, phosphorus, and potassium were 8.42 g·kg^−1^, 0.51 g·kg^−1^, 0.67 g·kg^−1^, and 7.21 g·kg^−1^, respectively. Meanwhile, the contents of available phosphorus, potassium, and nitrogen were 0.59 mg·kg^−1^; 51.01 mg·kg^−1^, and 13.17 mg·kg^−1^, respectively.

### 2.3. Root Sample Collection

A total of three plants were randomly selected with the same growth. The above-ground part was cut and the roots were removed completely after disassembling the flowerpot. The substrate that was attached to the roots was removed, and the root samples were collected from the plants with disinfected scissors. Each sample of 10 g was taken and placed into a sterile sealed bag, labeled, placed into a foam plastic box filled with ice, and transferred to the laboratory immediately. The sample was followed by a 1 min wash in 95% ethanol, a 14 min wash with agitation in sodium hypo-chlorite solution (1.4% active chlorine), and a 10 s wash in 95% ethanol, followed by 10 rinses in sterile water with agitation. The samples were then stored in a refrigerator at 4 °C for immediate analysis or at −80 °C for later use [28,29].

### 2.4. Analysis of Endophytic Microbial Composition

Extraction, PCR amplification, and sequencing of total DNA from root samples were performed by Majorbio Bio-Pharm Technology Co., Ltd. (Shanghai, China). The total DNA was extracted using the EZNA Soil DNA Kit (Omega Company, Irving, TX, USA), as per the manufacturer’s instructions. The DNA concentration and purity were detected using a NanoDrop2000 spectrophotometer (Thermo Fisher Scientific, Waltham, MA, USA), and the purity and quality of the genomic DNA were checked on a 1% agarose gel.

PCR products were extracted from 2% agarose gel, purified using an AxyPrep DNA gel extraction kit (Axygen Biosciences, Union City, CA, USA) according to the manufacturer’s instructions, and quantified using a Quantus fluorescence fluorimeter (Promega, Madison, WI, USA). For Illumina MiSeq sequencing, the PCR products of the same sample were purified using an AxyPrep DNA gel extraction kit (Axygen Biosciences, Union City, CA, USA), mixed, and detected using 2% agarose gel recovery. A Quantus fluorescence fluorometer (Promega, USA) was used to quantify the recovered products. Library construction was carried out using the NEXTFLEX^®^ Rapid DNA-Seq Kit [27]. For endophytic bacteria, the primer set 799F (5′-AACMGGATTAGATACCCKG-3′) and 1193R (5′-ACGTCATCCCCACCTTCC-3′) was used to amplify the V5–V7 hypervariable regions of the 16S rRNA gene [30]. Sequencing was performed using the Illumina MiSeq PE250 platform (Illumina, Inc., San Diego, CA, USA) For endophytic fungi, the primer set ITS1F (5′-CTTGGTCATTTAGAGAAGTAA-3′) and ITS2R (5′-GCTGCGTTCTTCATCGATGC-3′) was used [31]. Sequencing was performed using an Illumina MiSeq PE300 platform (Illumina, Inc., USA). Sequencing information of bacteria and fungi in root of tomatoes under different growth stages are shown in Appendix A.

UPARSE software [32] (http://drive5.com/uparse/, accessed on 13 January 2022), version 7.1, was used to cluster the operational taxon (OTU) with 97% similarity cut-off, and the chimeric sequence was identified and removed. The specific process and the OTU clustering steps were as follows: (1) extract non-redundant sequences from the optimized sequences to reduce redundant calculations in the intermediate analysis process (http://drive5.com/usearch/manual/dereplication.html, accessed on 13 January 2022); (2) remove singletons, which are sequences that do not have duplicates (http://drive5.com/usearch/manual/singletons.html, accessed on 13 January 2022); (3) perform OTU clustering on the non-redundant sequences (excluding singletons) at 97% similarity, and remove chimeras during the clustering process to obtain the representative sequences of OTUs; (4) map all optimized sequences to the OTU representative sequences, select sequences with more than 97% similarity to the representative sequences, and generate the OTU table. The software platform used was Uparse (version 7.0.1090 http://drive5.com/uparse/, accessed on 13 January 2022). We removed chloroplast and mitochondrial sequences and annotated them in all samples, and leveled all sample sequence numbers to that minimum sample number. After leveling, the average sequence of each sample reached 99.09%. Using a confidence threshold of 0.7, the classification of each OTU representative sequence was analyzed against a 16s rRNA gene database, Silva v138 (http://www.arb-silva.de, accessed on 13 January 2022) using RDP Classifier version 2.11 (http://RDP.CME.MSU.edu/, accessed on 13 January 2022). And the community composition of each sample was counted at different species classification levels.

The raw data were uploaded to the NCBI database for comparison. The data of the comparison database were taken from the following: bacterial from Silva 138 (http://www.arb-silva.de, accessed on 13 January 2022) and fungal from Unite 8.0 (http://unite.ut.ee/index.php, accessed on 13 January 2022).

### 2.5. Statistical Analysis

The experimental data were recorded in Excel 2019 for the ease of mathematical calculations. Statistical analyses were performed using SPSS statistics 22.0 (IBM Corp., Armonk, NY, USA), and Duncan’s multiple range test was used to compare means. Significance was based on a total of 999 Monte Carlo permutations. LEfSe (http://huttenhower.sph.harvard.edu/galaxy/root?tool_id=lefse_upload, accessed on 13 January 2022) was used to identify significantly different bacterial and fungal communities in different environmental samples. LEfSe carried out linear discriminant analysis (LDA) on samples according to taxonomic composition and different grouping conditions, and discovered the com-munities or species that demonstrated significant differences in sample division. Online data analysis was performed using the free online platform Majorbio Cloud Platform (http://www.majorbio.com, accessed on 13 January 2022) from Majorbio Bio-Pharm Technology Co., Ltd. (Shanghai, China). BugBase (https://bugbase.cs.umn.edu/index.html) is a microbiome analysis tool that normalizes OTUs by predicted 16S copy number and then predicts the microbial table using the provided pre-computed files, which we used for phenotyping, identifying high-level phenotypes present in microbiome samples, and performing phenotype predictions. The PICRUSt2 function prediction used the COG information stored in PICRUSt2 and corresponding to greengeneid to standardize the OTU abundance table; that is, to eliminate the influence of 16Smarkergene copies in the species genome. Then, the COG family information of the corresponding OTU was obtained through the greengeneid corresponding to each OTU, and the abundance was calculated. According to the information in the COG database, we analyzed the descriptive information and functional information of each COG in the egg wine database, so as to obtain the functional abundance spectrum (http://huttenhower.sph.harvard.edu/galaxy, accessed on 13 January 2022). FUNGuild (http://www.funguild.org/, accessed on 13 January 2022) classified fungal communities through a microecological guide, which was linked with functional guide classification to classify fungi functionally.

## 3. Results

### 3.1. Diversity of Endophytic Microorganisms in Tomato Roots at Different Growth Stages

As presented in Figure 1a–d, significant differences were found in endophytic bacterial and fungal diversity (Shannon) and richness (Ace) in tomato roots during the GY, GH, and GJ stages.

Partial least squares discriminant analysis (PLS-DA) was conducted to evaluate the similarity of the endophytic microbial communities at the OTU level. The results demonstrated that the endophytic microbial communities at GY, GH, and GJ stages were clustered separately, suggesting that the endophytic microbial composition in the roots of tomatoes could change during the different growth stages (Figure 1e–h).

Additionally, at the OTU and genus levels, the number of specific endophytic bacteria in tomato roots during the GJ stage, at 51, was higher than that in GY and GH. The number of unique endophytic fungi in tomato roots during the GJ stage was also higher than that of the GY and GH systems, with values of 291, 122, and 65, respectively (Figure 1i,j).

### 3.2. Composition of Endophytic Microorganisms in Tomato Roots at Different Growth Stages

As presented in Figure 2a, the number of dominant endophytic bacterial phyla (i.e., relative abundances greater than 1%) in the GY, GH, and GJ stages were three, four, and four, respectively.

First, the dominant endophytic bacterial phyla in the roots of tomatoes during the GY stage, from high to low, were *Proteobacteria* (77.14%), *Actinobacteriota* (18.75%), *Bacteroidota* (1.27%), and others (2.96%).

The dominant endophytic bacterial phyla in the roots of tomatoes during the GH stage were *Proteobacteria* (51.37%), *Actinobacteria* (26.40%), *Bacteroidetes* (15.45%), *Firmicutes* (15.45%), and others (2.96%).

The dominant endophytic bacterial phyla in the roots of tomatoes during the GJ stage were *Proteobacteria* (33.03%), *Actinobacteria* (51.03%), *Bacteroidetes* (9.44%), *Firmicutes* (3.25%), and others (3.25%).

The number of dominant endophytic fungal phyla (i.e., relative abundances greater than 1%) in the GY, GH, and GJ stages was five, two, and five, respectively (Figure 2b).

Firstly, *Olpidiomycota* (38.68%), *Chytridiomycota* (34.20%), *Ascomycota* (13.38%), *Mortierellomycota* (3.57%), and others (11.46%) were found to be the dominant endophytic fungal phyla in roots of tomatoes during the GY stage. Secondly, *Olpidiomycota* (96.52%) and *Ascomycota* (2.18%) were the only dominant endophytic fungal phyla in the roots of tomatoes during the GH stage. *Olpidiomycota* (52.84%), *Ascomycota* (40.15%), *Basidiomycota* (2.13%), *Glomeromycota* (2.79%), and others (4.69%) were the dominant endophytic fungal phyla in the tomato roots during the GJ stage. These results demonstrated that *Chytridiomycota* and *Mortierellomycota* were the unique dominant endophytic fungal phyla in the roots of tomatoes during GY. In contrast, *Basidiomycota* and *Glomeromycota* were the dominant endophytic fungal phyla in tomato roots during the GJ stage. However, *Chytridiomycota*, *Mortierellomycota*, *Basidiomycota*, and *Glomeromycota* were lost in tomato roots during GH.

As presented in Figure 2c, 25, 23, and 18 dominant endophytic bacterial genera (relative abundances are greater than 1%) were detected in the roots of tomatoes during the GY, GH, and GJ stages, respectively.

Firstly, *Massilia* (13.04%), *Asticcacaulis* (8.79%), *Rhodanobacter* (6.04%), *Allorhizobium*-*Neorhizobium*-*Pararhizobium*-*Rhizobium* (4.46%), *Leifsonia* (4.04%), *Sphingobium* (3.66%), *Ensifer* (3.46%), *Streptomyces* (3.22%), *Ralstonia* (3.17%), *Burkholderia*-*Caballeronia*-*Paraburkholderia* (2.72%), *Devosia* (2.30%), *Nocardioides* (2.11%), *Phycicoccus* (1.98%), *Acidovorax* (1.73%), *Sphingomonas* (1.66%), *Dyella* (1.64%), *Pseudomonas* (1.30%), *Actinospica* (1.23%), *Flavobacterium* (1.19%), *Bradyrhizobium* (1.07%), and others (40.39%) were found to be the dominant endophytic bacterial genera in the roots of tomatoes during the GY stage.

*Flavobacterium* (13.46%), *Devosia* (7.20%), *Acidovorax* (6.74%), *Streptomyces* (3.90%), *Allorhizobium*-*Neorhizobium*-*Pararhizobium*-*Rhizobium* (3.86%), *Sphingobium* (2.78%), *Nocardioides* (2.45%), *Ensifer* (2.43%), *Thermomonas* (1.93%), *Leifsonia* (1.86%), *Massilia* (1.65%), *Pseudomonas* (1.64%), *Asticcacaulis* (1.56%), *Dyella* (1.52%), *Phycicoccus* (1.38%), *Rhodanobacter* (1.14%), *Bradyrhizobium* (1.07%), *Actinomadura* (1.02%), *Bacillus* (1.02%), and others (35.24%) were determined to be the dominant endophytic bacterial genera in roots of tomatoes during the GH stage.

Furthermore, *Flavobacterium* (7.52%), *Nocardioides* (4.63%), *Lechevalieria* (4.14%), *Streptomyces* (3.72%), *Devosia* (2.95%), *Acidovorax* (2.85%), *Actinoplanes* (2.52%), *Thermomonas* (2.42%), *Dyella* (1.94%), *Allorhizobium*-*Neorhizobium*-*Pararhizobium*-*Rhizobium* (1.75%), *Massilia* (1.66%), *Pseudomonas* (1.36%), *Sphingobium* (1.25%), *Sphingomonas* (1.07%), and others (53.74%) were found to be the dominant endophytic bacterial genera in the roots of tomatoes during the GJ stage.

The results discussed above demonstrated that *Massilia*, *Ralstonia*, *Burkholderia*-*Caballeronia*-*Paraburkholderia*, and *Actinospica* were the special endophytic bacterial genera in the roots of tomatoes during the GY stage. Meanwhile, *Actinomadura* and *Bacillus* were found to be the unique endophytic bacterial genera in the roots of tomatoes during the GH stage.

*Actinoplanes* and *Lechevalieria* were determined to be the special endophytic bacterial genera in the roots of tomatoes during the GJ stage. Additionally, the number of dominant endophytic fungal genera (i.e., those with relative abundances greater than 1%) in the GY, GH, and GJ stages were determined to be nine, two, and twelve, respectively (Figure 2d).

Firstly, *Olpidium* (38.68%), *Phoma* (6.48%), *Mortierella* (3.55%), *Poaceascoma* (1.23%), *Powellomyces* (1.08%), and others (46.95%) were found to be the dominant endophytic fungal genera in the roots of tomatoes during the GY stage.

*Olpidium* (96.52%) and others (1.23%) were determined to be the dominant endophytic fungal genera in the tomato roots during the GH stage.

Thirdly, *Olpidium* (52.84%), *Plectosphaerella* (13.21%), *Fusarium* (3.99%), *Zopfiella* (2.37%), *Poaceascoma* (1.18%), *Gibberella* (1.12%), and others (24.02%) were found to be the dominant endophytic fungal genera in the roots of tomatoes at the GJ stage. These results also revealed that *Wallemia* (35.35%), *Myrmecridium* (1.07%), and *Sodiomyces* (1.21%) were the unique endophytic fungal genera in the roots of tomatoes during the GY stage. Meanwhile, *Plectosphaerella* (13.21%), *Fusarium* (3.99%), *Zopfiella* (2.37%), and *Gibberella* (1.12%) were found to be the special endophytic fungal genera in the roots of tomatoes during the GJ stage.

Additionally, the LEfSe analysis of endophytic microorganisms in the roots of tomatoes during the GY, GH, and GJ stages demonstrated significant differences (LDA > 3.0) in their cladogram structures.

As presented in Figure 3a, at the phylum and genus levels, one phylum of bacteria and seven genera of bacteria were significantly enriched in the GY treatment—*Proteobacteria*, *Asticcacaulis*, *Leifsonia*, *Actinospica*, *Terracidiphilus*, *Dokdonella*, and *Candidatus Solibacter*, respectively.

In comparison to the GY stage, *Promicromonospora* was enriched as an endophytic bacterial genus in the GH stage.

At the phylum and genus levels, one phylum of bacteria and six genera of bacteria were significantly enriched in the GJ treatment—Actinobacteriota, *Lechevalieria*, *Amycolatopsis*, *Ferrovibrionaccae*, *Lysobacter*, and *Methylophilus*.

Moreover, the number of dominant soil fungal groups in the GY system was similar to that in the GJ system. For example, *Cyberlindnera* was enriched in the GH stage, whereas *Ascomycota*, *Zopfiella*, *Funneliformis*, *Oliveonia*, and *Monosporascus* were enriched in the GH stage (Figure 3b).

### 3.3. Function Prediction of Endophytic Microorganisms in Tomato Root System at Different Growth Stages

The results of the BugBase analysis demonstrated that the abundance of endophytic bacteria, particularly Gram-positive, Gram-negative, aerobic, and potentially pathogenic bacteria, were all significantly different in the roots of tomatoes in the GY, GH, and GJ stages (Figure 4a).

PICRUSt2, FUNGuild, Wilcoxon rank-sum, and Student’s *t*-tests were all carried out to evaluate the functions of the endophytic bacterial and fungal communities in the roots of tomatoes during the GY, GH, and GJ stages (*p* < 0.05). The results presented no significant differences in endophytic bacteria (Figure 4b) and fungi (Figure 4c) in the tomato roots at the GY, GH, and GJ stages.

## 4. Discussion

Hormones play an important role in plant growth and development, such as abscisic acid and ethylene [33]. For example, abscisic acid is instrumental in controlling essential genes in tomatoes and contributes significantly to ethylene production, particularly during the blooming phase [34]. Ethylene synthesis, perception, and signaling are also vital for fruit ripening [35], with ethylene production markedly increasing during specific phases of plant growth, such as fertilization, maturity, aging, and shedding, and under biotic or abiotic stress [36]. Ethylene also plays a key role in the accumulation of carotenoids during fruit ripening, and gibberellin is negatively correlated with the maturation of tomato fruits [37].

Moreover, studies have demonstrated a close relationship between the generation of endogenous plant hormones and microorganisms. For example, the sources of several plant hormones, including abscisic acid [38], ethylene [39], gibberellin [40], and auxin were derived from bacteria and fungi, such as *Ralstonia* producing auxin and ethylene [41], while also simulating the production of jasmonic acid [42]. Moreover, the rapid increase in the levels of abscisic acid during fruit development was positively correlated with *Actinomycetes* and negatively correlated with Actinobacteria [43], with Actinobacteria predominating in the tomato seedling stage [8].

Additionally, *Massilia* has been proven to be a successful colonizer in the early stages of plant growth [9]. Meanwhile, *Sphingomonas* can inhibit disease development and pathogen growth [44], possess bacteriostatic activity [45], and encode anti-host reactive oxygen species, such as iron peroxidase and arginase [46]. In this study, *Massilia*, *Sphingomonas*, and *Leifsonia* were found to be the dominant endophytic bacterial genera in tomato roots during the GY stage.

Moreover, *Bacillus* was detected as the dominant endophytic bacterial genus, and *Streptomyces*, *Flavobacterium*, *Pseudomonas*, *Acidovorax*, and *Actinomadura* were found to be the dominant endophytic bacterial genera in tomato roots during GH. Previous research has demonstrated that *Bacillus* induces the accumulation of abscisic acid [47] and gibberellin [48]. *Streptomyces* can improve nutrient supply, carbohydrate accumulation, and the dry weight of tomato plants. They also improved stress resistance [49]. *Streptomyces* promoted tomato growth [50]. Furthermore, *Bacillus* and *Streptomyces* promoted plant growth through phosphorus solubilization and mobilization [51]. *Flavobacterium* is known for its ability to degrade complex organic compounds, such as building resistance to plant pathogens, producing plant hormones [52], and significantly increasing plant growth, yield, and protein content under different environmental conditions; *Acidovorax* promotes plant growth and improves nutrient utilization [53]; and *Actinomadura* is positively correlated with available phosphorus [54].

*Nocardioides* promotes plant growth related to nitrogen fixation [55] and *Fusarium* produces ethylene [56] and abscisic acid [57]. Actinobacteria was found to be the dominant endophytic bacterial phylum in tomato roots at the GJ stage in this study, while *Nocardioides* and *Fusarium* were determined to be the dominant bacterial and fungal genera, respectively.

The results of this study indicate that auxin-producing microorganisms were primarily enriched as the dominant endophytes in tomato roots during the GY stage. In contrast, stress-resistant microorganisms primarily survived as the dominant endophytes in tomato roots during the flower bud differentiation stage. That nutrient cycling promoted microorganisms, chiefly colonized as the dominant endophytes in tomato roots during the GJ stage.

## 5. Conclusions

The endophytic microbial composition was significantly altered in tomato roots during different growth stages. Auxin-producing microorganisms, such as *Massilia*, *Sphingomonas*, and *Leifsonia*, were primarily enriched as the dominant endophytes in tomato roots during the seedling (GY) stage; stress-resistant microorganisms and plant growth-promoting endophytes, such as *Streptomyces*, *Flavobacterium*, *Pseudomonas*, *Acidovorax*, and *Actinomadura*, were found to be the dominant endophytic bacterial genera in tomato roots at the flower bud differentiation (GH) stage; *Nocardioides* and *Fusarium* were considered to be the dominant endophytes of nutrient cycling in tomato roots during the fruiting (GJ) stage. These results indicated that the enrichment of exact endophytes in tomato roots during different growth stages could assemble diverse community structures, with functions relating to plant growth promotion, stress resistance, and nutrient cycling improvement, respectively.

## Figures and Tables

**Figure 1 microorganisms-12-01251-f001:**
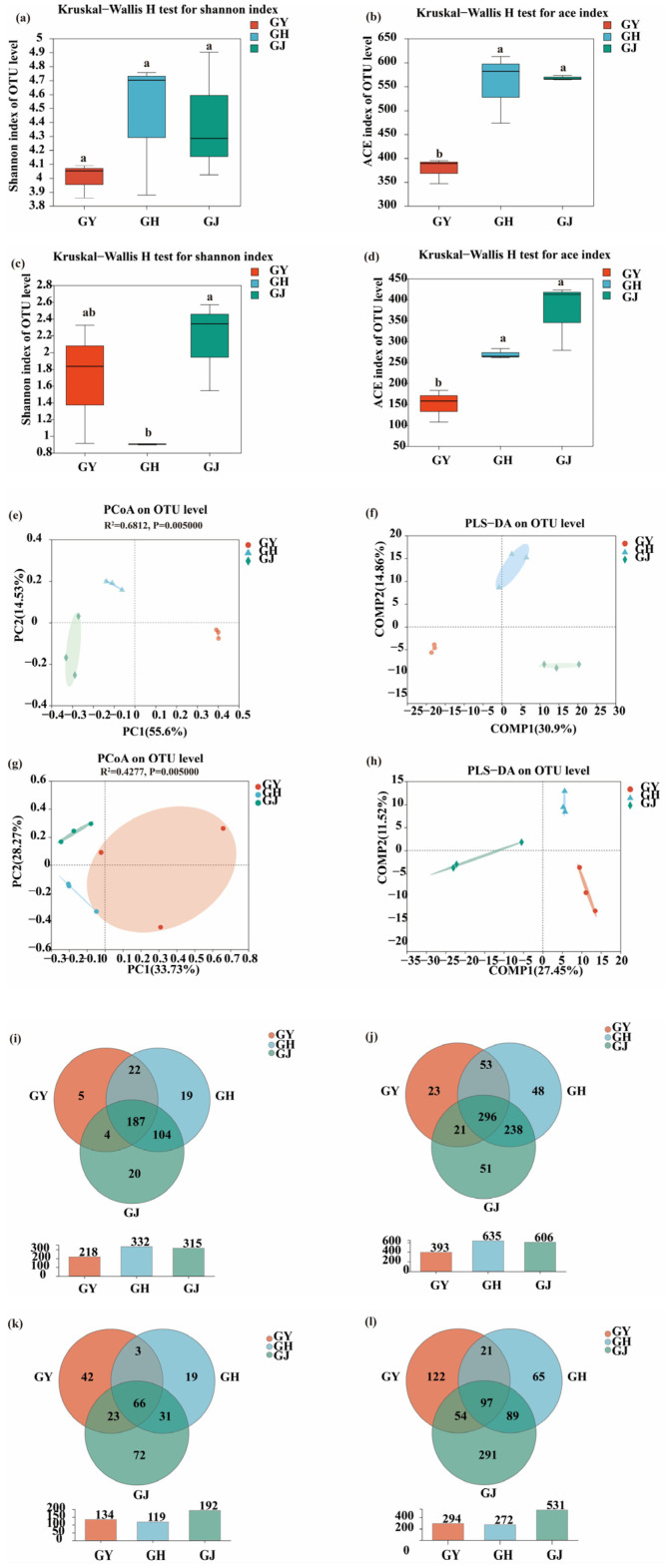
Comparison of endophytic microbiota structures in tomato roots at a similarity level of 97% between GY, GH, and GJ treatments ((operational taxonomic unit) OTU level). (**a**) The Shannon index indicates endophytic bacterial diversity. (**b**) The Ace index indicates endophytic bacterial richness. (**c**) The Shannon index indicates endophytic fungal diversity. (**d**) The Ace index indicates endophytic fungal richness. (**e**) PCoA score plot of endophytic bacteria communities. (**f**) PLS-DA score plot of endophytic fungi communities. (**g**) PCoA score plot of endophytic bacteria communities. (**h**) PLS-DA score plot of endophytic fungi communities. (**i**) Venn diagram analyses of endophytic bacteria at the genus level. (**j**) Venn diagram analyses of endophytic bacteria at the OTU level. (**k**) Venn diagram analyses of endophytic fungi at the genus level. (**l**) Venn diagram analyses of endophytic fungi at the OTU level. GY: seedling stage; GH: flower bud differentiation stage; GJ: fruiting period. The same letters on the bars within a figure indicate no significant differences in the mean ranks among treatments at *p* < 0.05.

**Figure 2 microorganisms-12-01251-f002:**
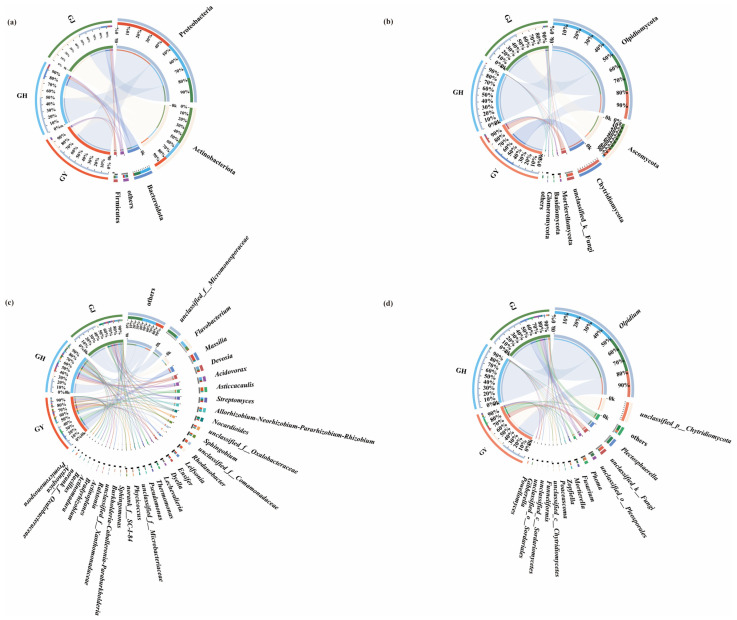
(**a**) Compositions of endophytic bacterial communities at the phylum level; (**b**) compositions of endophytic fungal communities at the phylum level; (**c**) compositions of endophytic bacterial communities at the genus level; (**d**) compositions of endophytic fungal communities at the genus level under the GY, GH, and GJ treatments. GY: seedling stage; GH: flower bud differentiation stage; GJ: fruiting stage.

**Figure 3 microorganisms-12-01251-f003:**
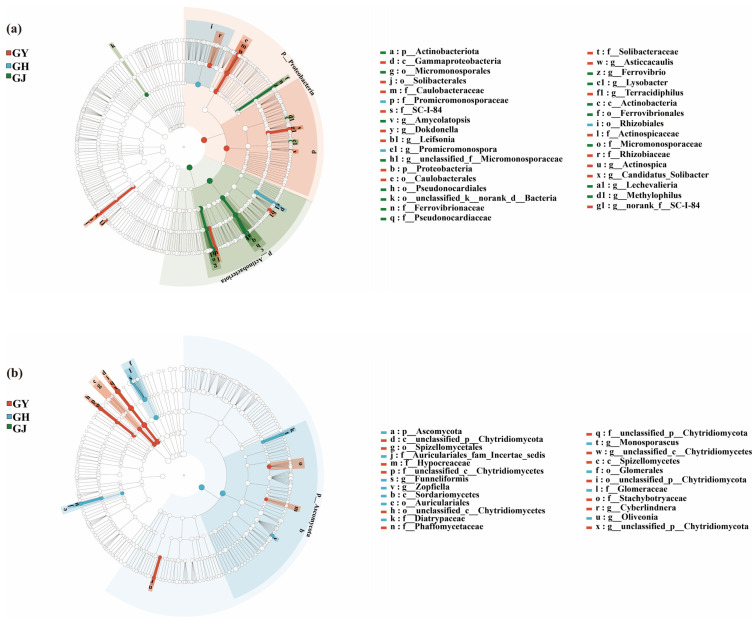
LEfSe analysis of endophytic bacteria (**a**) and fungi (**b**) in tomato roots under GY, GH, and GJ stages. GY: seedling stage; GH: flower bud differentiation stage; GJ: fruiting stage.

**Figure 4 microorganisms-12-01251-f004:**
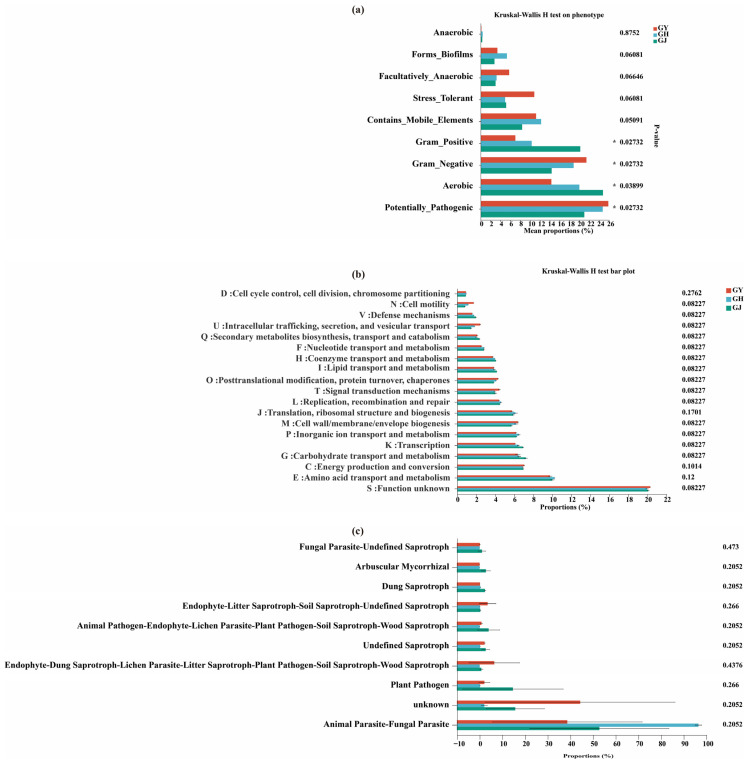
Relative abundance of BugBase (**a**), COG (**b**), and FUNGuild (**c**) inferred the endophytic microbial functions in tomato roots under GY, flower bud GH, and GJ stages. GY: seedling stage; GH: flower bud differentiation stage; GJ: fruiting period. * *p* ≤ 0.05.

## Data Availability

Raw data for endophytic bacterial and fungal sequencing were deposited in the NCBI Sequence Read Archive (SRA) database under accession number PRJNA1088469 (accessed on 16 March 2024) and PRJNA1088731 (accessed on 20 March 2024), respectively.

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
