# Peer review of "Endophytic Microorganisms in Tomato Roots, Changes in the Structure and Function of the Community at Different Growing Stages"

_microorganisms, 2024, doi:10.3390/microorganisms12061251_

Round 1

Reviewer 1 Report

Comments and Suggestions for Authors

The manuscript describes the use of metagenomic approaches to determine the changes in the microbial community of endophytes in roots of tomato. Three growth stages were evaluated, . GY: seedling stage; GH: flower bud differentiation stage. GJ: fruiting stage; bacterial and fungal community structure and diversity was different in each growth stages. Authors propose that the enrichment of endophytes in tomato roots during different growth stages contributes to the promotion of plant growth, stress resistance, and nutrient cycling improvement.

Several aspects must be corrected in the manuscript

Commentaries

The numbering of lines in the manuscript is necessary to facilitate revision

In keywords use italics for “Solanum”

In the introduction, third paragraph, check redaction in “For instance, core microbiomes, such as Firmicutes and Proteobacteria [11], Arabidopsis [12], and Boechera stricta [13] could be altered.[14]” it is not clear.

In the introduction, third paragraph, check format in “altered.[14]”

In the introduction, last paragraph, complement the redaction with possible applications of the knowledge derived of the characterization of the endophytes communities in roots of tomato plants in different growth stages.

In Materials and Methods, first paragraph, “22.13°C” could be “22.1°C”

In “Root Sample Collection” section fourth line, “disinfection scissors” could be “disinfected scissors”

In “Analysis of Endophytic Microbial Composition” section, second paragraph, add a period in “(Illumina, Inc., USA) For”

In “Analysis of Endophytic Microbial Composition” section, second paragraph, add a space in “ITS1F(5’-CTTGGTCATTTAGAGAAGTAA-3’) and ITS2R(5’-GCTGCGTTCTTCATCGATGC-3’)”

Check format in “Unite 8.0, (http://unite.ut.ee/index.php)”, eliminate comma

Check redaction in “Additionally, the number of specific endophytic bacteria in tomato roots during the GJ stage had 51 was higher than that in GY and GH at the OTU and genus levels”

In the Figure 1 caption, “Ace” could be “ACE”

In “Composition of Endophytic Microorganisms in Tomato Roots at Different Growth Stages” section, eliminate “in the roots of tomatoes during the GY stage” and “in the roots of tomatoes during the GH stage”

“As presented in Figures 2c” could be “As presented in Figure 2c”

In page 6, use italics in the descriptions of all bacteria and fungi genera identified

I n page 7, check format in “CandidatusSolibacter

In Figure 3 caption, the following information “GY: seedling stage; GH: flower bud differentiation stage. GJ: fruiting stage” is showed, add these information in al figures to enhance comprehension of the results

In page 8, “gram-positive and gram-negative” must be “Gram-positive and Gram-negative”

In figure 4 caption check “*, 0.01 p = 0.05; **, 0.001, p = 0.01; ***, p = 0.001.” it is confusing

The conclusions are too general and descriptive, like the description in the results section. Restructure the conclusion, with aspects such as the relevance of the observed changes in the microbial community, which benefits could be expected from the presence of the main bacterial and fungal genera identified, and which strategies to improve growth, productivity, or stress tolerance can be designed from the information generated in the study.

Author Response

Dear Editors and Reviewers:

Thank you for your letter and the reviewers’ comments concerning our manuscript entitled “Characteristics of endophytic microbial community structures in tomato roots under different growing stages” (Submission ID: microorganisms-2996270). Those comments are all valuable and very helpful for revising and improving our paper, as well as the important guiding significance tour researches. We have read comments carefully and have made correction which we hope meet with approval. Revised portion are marked in Green in the paper. The main corrections in the paper and the responds to the reviewer’s comments are as flowing:

Reviewers1

The manuscript describes the use of metagenomic approaches to determine the changes in the microbial community of endophytes in roots of tomato. Three growth stages were evaluated, GY: seedling stage; GH: flower bud differentiation stage. GJ: fruiting stage; bacterial and fungal community structure and diversity was different in each growth stages. Authors propose that the enrichment of endophytes in tomato roots during different growth stages contributes to the promotion of plant growth, stress resistance, and nutrient cycling improvement.

Several aspects must be corrected in the manuscript

Commentaries

My comments are below:

1.The numbering of lines in the manuscript is necessary to facilitate revision

Response: The authors have revised this point in manuscript according to the Reviewer’s suggestion.

2.In keywords use italics for “Solanum”

Response: The authors have revised this point in manuscript according to the Reviewer’s suggestion(L27).

3.In the introduction, third paragraph, check redaction in “For instance, core microbiomes, such as Firmicutes and Proteobacteria [11], Arabidopsis [12], and Boechera stricta [13] could be altered.[14]” it is not clear.

Response: The authors have revised this point in manuscript according to the Reviewer’s suggestion(L49-50).

4.In the introduction, third paragraph, check format in “altered.[14]”

Response: The authors have revised this point in manuscript according to the Reviewer’s suggestion(L49).

5.In the introduction, last paragraph, complement the redaction with possible applications of the knowledge derived of the characterization of the endophytes communities in roots of tomato plants in different growth stages.

Response:

 The authors have found and added the refernces below in this manuscript. i.e. Jiao et al., 2024[23] to support the statements and revised this point in manuscript according to the Reviewer’s suggestion.

6.In Materials and Methods, first paragraph, “22.13°C” could be “22.1°C”

Response: The authors have revised this point in manuscript according to the Reviewer’s suggestion(L85).

7.In “Root Sample Collection” section fourth line, “disinfection scissors” could be “disinfected scissors”

Response: The authors have revised this point in manuscript according to the Reviewer’s suggestion(L105).

8.In “Analysis of Endophytic Microbial Composition” section, second paragraph, add a period in “(Illumina, Inc., USA) For”

Response: The authors have revised this point in manuscript according to the Reviewer’s suggestion(L124).

9.In “Analysis of Endophytic Microbial Composition” section, second paragraph, add a space in “ITS1F(5’-CTTGGTCATTTAGAGAAGTAA-3’) and ITS2R(5’-GCTGCGTTCTTCATCGATGC-3’)”

Response: The authors have revised this point in manuscript according to the Reviewer’s suggestion(L125).

10.Check format in “Unite 8.0, (http://unite.ut.ee/index.php)”, eliminate comma

Response: The authors have revised this point in manuscript according to the Reviewer’s suggestion(L144).

11.Check redaction in “Additionally, the number of specific endophytic bacteria in tomato roots during the GJ stage had 51 was higher than that in GY and GH at the OTU and genus levels”

Response: The authors have revised this point in manuscript according to the Reviewer’s suggestion(L169).

12.In the Figure 1 caption, “Ace” could be “ACE”

Response: The authors have revised this point in manuscript according to the Reviewer’s suggestion(L173).

13.In “Composition of Endophytic Microorganisms in Tomato Roots at Different Growth Stages” section, eliminate “in the roots of tomatoes during the GY stage” and “in the roots of tomatoes during the GH stage”

Response: The authors have revised this point in manuscript according to the Reviewer’s suggestion(L190).

14.“As presented in Figures 2c” could be “As presented in Figure 2c”

Response: The authors have revised this point in manuscript according to the Reviewer’s suggestion(L229).

15.In page 6, use italics in the descriptions of all bacteria and fungi genera identified

Response: The authors have revised this point in manuscript according to the Reviewer’s suggestion.

16.I n page 7, check format in “CandidatusSolibacter

Response: The authors have revised this point in manuscript according to the Reviewer’s suggestion(L268).

17.In Figure 3 caption, the following information “GY: seedling stage; GH: flower bud differentiation stage. GJ: fruiting stage” is showed, add these information in all figures to enhance comprehension of the results

Response: The authors have revised this point in manuscript according to the Reviewer’s suggestion(L281).

18.In page 8, “gram-positive and gram-negative” must be “Gram-positive and Gram-negative”

Response: The authors have revised this point in manuscript according to the Reviewer’s suggestion(L286).

19.In figure 4 caption check “*, 0.01 p = 0.05; **, 0.001, p = 0.01; ***, p = 0.001.” it is confusing

Response: The authors have revised this point in manuscript according to the Reviewer’s suggestion(L297).

The conclusions are too general and descriptive, like the description in the results section. Restructure the conclusion, with aspects such as the relevance of the observed changes in the microbial community, which benefits could be expected from the presence of the main bacterial and fungal genera identified, and which strategies to improve growth, productivity, or stress tolerance can be designed from the information generated in the study.

Response: Response: The authors have revised this point in manuscript according to the Reviewer’s suggestion(L347).

Reviewer 2 Report

Comments and Suggestions for Authors

The paper addresses an interesting topic, namely the composition of microorganisms inhabiting tomato roots at different stages of plant growth. The authors employed correct methods to determine the species composition of organisms and presented the research results in an interesting and understandable manner. However, a weak aspect of the paper is the brief description of the conditions under which the tomatoes were cultivated. The authors should precisely specify: 1. Whether the cultivations were conducted outdoors or in a growth chamber. 2. Where they sourced the soil for cultivation. 3. Whether the soil was covered with any vegetation or perhaps crops. Including a photo of the cultivation setup, for example, as supplementary materials, would greatly enhance the study. I believe this is an intriguing piece of work, but it should be regarded more as information regarding the specific conditions of tomato cultivation employed by the authors, on a specific type of soil, rather than as a model for interpreting the colonization of tomato root microorganisms in general.

All figures should be made more readable. A good option might be to separate them into larger parts, such as placing diagrams a and b from Figure 1 on one line, and placing diagrams c and d below.

Author Response

Thank you for your letter and the reviewers’ comments concerning our manuscript entitled “Characteristics of endophytic microbial community structures in tomato roots under different growing stages” (Submission ID: microorganisms-2996270). Those comments are all valuable and very helpful for revising and improving our paper, as well as the important guiding significance tour researches. We have read comments carefully and have made correction which we hope meet with approval. Revised portion are marked in Green in the paper. The main corrections in the paper and the responds to the reviewer’s comments are as flowing:

Reviewers2

The paper addresses an interesting topic, namely the composition of microorganisms inhabiting tomato roots at different stages of plant growth. The authors employed correct methods to determine the species composition of organisms and presented the research results in an interesting and understandable manner. However, a weak aspect of the paper is the brief description of the conditions under which the tomatoes were cultivated. The authors should precisely specify:

1.Whether the cultivations were conducted outdoors or in a growth chamber.

Response: All samples of tomatoes were collected together after one month of growth under identical conditions in the greenhouse. The authors have added this point in manuscript(L82-84).

  1. Where they sourced the soil for cultivation.

Response: The planting soil comes from the experimental soil of vegetable base of Agricultural College of Guangxi University, and the soil type is acidic red loam.

  1. Whether the soil was covered with any vegetation or perhaps crops. Including a photo of the cultivation setup, for example, as supplementary materials, would greatly enhance the study. I believe this is an intriguing piece of work, but it should be regarded more as information regarding the specific conditions of tomato cultivation employed by the authors, on a specific type of soil, rather than as a model for interpreting the colonization of tomato root microorganisms in general.

Response: Thank you for your good comments. The soil type of the study area was acid red loam. Before planting, there are no plants covered on the soil surface.

All figures should be made more readable. A good option might be to separate them into larger parts, such as placing diagrams a and b from Figure 1 on one line, and placing diagrams c and d below.

Response: Thank you for your good comments. The authors have revised this point in manuscript according to the Reviewer’s suggestion(L173).

Reviewer 3 Report

Comments and Suggestions for Authors

The manuscript with the title “Characteristics of Endophytic Microbial Community Structures 2 in Tomato Roots under Different Growing Stages” is describing the endophytic bacterial and fungal diversity in the roots of tomato plants at three growing stages. Overall, the manuscript is of general interest and mostly easily enough to follow.

General comments

The manuscript relies solely on amplicon sequencing results, no metabolomics or plant stress factors/plant hormones were determined at the three sampling stages. This is limiting the value of the manuscript and make it a one-sided read that is heavily relying on speculations based on the presence of bacterial and fungal taxa alone.

The manuscript is leaving out major steps in the data generation and data analysis and therefore the current version of the manuscript can not be fully evaluated. What was the sequencing depth, how much co-amplification of plant DNA took place, was the resulting sequence depth sufficient for unbiased analysis? How was the sequencing conducted and where, read analysis pipeline needs to be more thoroughly described either in main document or as supplementary section.

Results are vey descriptive and repetitive. The discussion is highly speculative as abundances of bacterial and fungal taxa is the sole measurement in the presented study.

The Conclusions is partly a summary and partly a speculation that is in my point of view not valid and needs to be revised.

Specific comments

Title - Title could more clearly highlight the findings then being purely descriptive.

Abstract

13 What is meant with tried?

16 particular instead of special?

18-9 How was that measured? Abstract contains no method background

20 How was “functional” determined?

Keywords – Genus name in italics.

Introduction

28 Provide example of how this has been done.

29 Specify what anti-aging method is.

31 Bacteria and/or fungi?

44 in plants such as?

61 Please consider other words to describe  extended plant vitality. Overall, the introduction is leading towards a path but no measurements were taken from the plant itself. Consider revising.

72 Please formulate a testable hypothesis for the described work.

Materials and Methods

81 Soil type?

81-5 Methods used to describe values? Olsen P as available P? Please either describe method of analysis or cite.

88 Age is not a treatment, please revise.

95 Not introduced above, is soil meant?

96 disinfected

101 Method described to effectively remove rhizoplane microbes? Cite or describe.

123 Why wasn't the QIIME 2 classifier used? The RDP classifier has been reported to be quite inaccurate. Likewise, why OTUs and not ASVs?

124 Please elaborate

103-27 Details on data analysis are missing. How many of the reads were of plant origin? How many reads were obtained in total? Was the sampling depth sufficient for analysis? Rarefied? A lot of basic information is missing in order to make a clear decision on the quality of the data provided.

139 Were both used and if so what were they exactly used for? Expand and explain.

Results

143 According to the figures this is not the case. All have letter A above the columns.

150 Unclear whether data set is weighed or unweighed.

Figure 1 Labelling very small so that reading is difficult even at 200%. Anosim or similar comparisons to find out if the separation is significant?

189 and throughout manuscript unclassified etc., this is not a phylum, please remove.

205-223 Genus names in italics.

205pp Unclassified is not a genus. Remove taxa that are not classified at taxon level. The whole section is largely a written form of figure 2 and therefore quite repetitive.

Discussion

297 hormones play

297-305 It would have been useful to measure some of these metabolomes/enzymes in parallel to the endophyte analysis.

338-43 The study has limited value as no plant metabolomes were measured and therefore, the presence of endophytes is purely discussed on the basis of other studies’ findings. I would have expected to have at least some key measures of the plant root to be included in the analysis.

At the moment, the study is quite one sided and limited. This is also evident from the short and limited discussion.

Conclusions

345-57 This is not a conclusion

357-9 This is pure speculation at this point as the conditions of the plants were not measured other than the growth stage.

Data availability

371 Provide accession numbers for each growth stage.

Comments on the Quality of English Language

Please see specific comments

Author Response

Dear Editors and Reviewers:
Thank you for your letter and the reviewers’ comments concerning our manuscript entitled “Characteristics of endophytic microbial community structures in tomato roots under different growing stages” (Submission ID: microorganisms-2996270). Those comments are all valuable and very helpful for revising and improving our paper, as well as the important guiding significance tour researches. We have read comments carefully and have made correction which we hope meet with approval. Revised portion are marked in Green in the paper. The main corrections in the paper and the responds to the reviewer’s comments are as flowing:

Reviewers3
General comments
The manuscript relies solely on amplicon sequencing results, no metabolomics or plant stress factors/plant hormones were determined at the three sampling stages. This is limiting the value of the manuscript and make it a one-sided read that is heavily relying on speculations based on the presence of bacterial and fungal taxa alone.
The manuscript is leaving out major steps in the data generation and data analysis and therefore the current version of the manuscript can not be fully evaluated. What was the sequencing depth, how much co-amplification of plant DNA took place, was the resulting sequence depth sufficient for unbiased analysis? How was the sequencing conducted and where, read analysis pipeline needs to be more thoroughly described either in main document or as supplementary section.
Results are vey descriptive and repetitive. The discussion is highly speculative as abundances of bacterial and fungal taxa is the sole measurement in the presented study.
The Conclusions is partly a summary and partly a speculation that is in my point of view not valid and needs to be revised.

Response: The authors have revised this point in manuscript according to the Reviewer’s suggestion

Specific comments
Title - Title could more clearly highlight the findings then being purely descriptive.
Response:The authors have revised this point in manuscript according to the Reviewer’s suggestion(L2)

Abstract

13 What is meant with tried?
Response: The authors have revised this point in manuscript according to the Reviewer’s suggestion(L13)

16 particular instead of special?

Response: The authors have revised this point in manuscript according to the Reviewer’s suggestion(L22).

18-9 How was that measured? Abstract contains no method background

Response: The authors have revised this point in manuscript according to the Reviewer’s suggestion(L16-18).

20 How was “functional” determined?

Response: We use PICRUSt2 and FUNGuild to predict the functional information of microbial communities in environmental samples, and classify and analyze bacterial and fungal communities through functional composition and abundance, so as to further understand some potential microbial functional traits in the process of environmental change.

Keywords – Genus name in italics.
Response: The authors have revised this point in manuscript according to the Reviewer’s suggestion(L27).

Introduction

28 Provide example of how this has been done.
Response: The authors have found and added the refernces below in this manuscript (Mi et al 2022) to support the statements and revised this point in manuscript according to the Reviewer’s suggestion (L34).
Mi, J.; Vallarino, J. G.; PetÅ™ík, I.; Novák, O.; Correa, S. M.; Chodasiewicz, M.; Havaux, M.; Rodriguez-Concepcion, M.; Al-Babili, S.; Fernie, A. R.; Skirycz, A.; Moreno, J. C. A Manipulation of Carotenoid Metabolism Influence Biomass Partitioning and Fitness in Tomato. Metabolic Engineering, 2022, 70, 166–180. https://doi.org/10.1016/j.ymben.2022.01.004.

29 Specify what anti-aging method is.
Response: The authors have found and added the refernces below in this manuscript (Mi et al 2022) to support the statements and revised this point in manuscript according to the Reviewer’s suggestion (L34)
Mi, J.; Vallarino, J. G.; PetÅ™ík, I.; Novák, O.; Correa, S. M.; Chodasiewicz, M.; Havaux, M.; Rodriguez-Concepcion, M.; Al-Babili, S.; Fernie, A. R.; Skirycz, A.; Moreno, J. C. A Manipulation of Carotenoid Metabolism Influence Biomass Partitioning and Fitness in Tomato. Metabolic Engineering, 2022, 70, 166–180. https://doi.org/10.1016/j.ymben.2022.01.004.

31 Bacteria and/or fungi?
Response: The authors have revised this point in manuscript according to the Reviewer’s suggestion.(L36)

44 in plants such as?

Response: The authors have revised this point in manuscript according to the Reviewer’s suggestion.

61 Please consider other words to describe extended plant vitality. Overall, the introduction is leading towards a path but no measurements were taken from the plant itself. Consider revising.

Response: The authors have revised this point in manuscript according to the Reviewer’s suggestion(L65).

72 Please formulate a testable hypothesis for the described work.

Response: The authors have revised this point in manuscript according to the Reviewer’s suggestion(L79).

Materials and Methods

81 Soil type?

Response: The soil type of the study area was acid red loam . Added at Materials and Methods(L88).

81-5 Methods used to describe values? Olsen P as available P? Please either describe method of analysis or cite.

Response: The authors have revised this point in manuscript according to the Reviewer’s suggestion(L92). Soil pH was measured using a pH meter (soil–water ratio 1:2.5). Total nitrogen (TN) wasdetermined using the semimicro-Kjeldahl method. Total phosphorus (TP) was determined by the alkali fusion-molybdenum anti-colorimetric method. Total potassium (TK)was determined by alkali fusion-flame spectrophotometry. Available P was determined using acid-fluoride solutions method, and available N, K were determined alkali diffusion method and flame photometry respectively.

88 Age is not a treatment, please revise.

Response: The authors have revised this point in manuscript according to the Reviewer’s suggestion(L93).

95 Not introduced above, is soil meant?

Response: The authors have revised this point in manuscript according to the Reviewer’s suggestion(L93).

96 disinfected

Response: The authors have revised this point in manuscript according to the Reviewer’s suggestion.(L128)

101 Method described to effectively remove rhizoplane microbes? Cite or describe.

Response: The author refers to the following references in this article to support these statements.

Emami, S.; Alikhani, H.A.; Pourbabaei, A.A.; Etesami, H.; Sarmadian, F.; Motessharezadeh, B. Effect of Rhizospheric and Endophytic Bacteria with Multiple Plant Growth Promoting Traits on Wheat Growth. Environ Sci Pollut Res. 2019, 26(19), 19804-19813. doi:10.1007/s11356-019-05284-x.

Beckers, B.; Op De Beeck, M.; Weyens, N.; Boerjan, W.; Vangronsveld, J. Structural Variability and Niche Differentiation in the Rhizosphere and Endosphere Bacterial Microbiome of Field-Grown Poplar Trees. Microbiome, 2017, 5. https://doi.org/10.1186/s40168-017-0241-2.

123 Why wasn't the QIIME 2 classifier used? The RDP classifier has been reported to be quite inaccurate. Likewise, why OTUs and not ASVs?

Response: Thank you for your good comments. The ASV process uses a denoising method that removes information from individual sequences, which reduces the number of valid sequences compared to the OTU process and may filter out some valid sequences.

124 Please elaborate

Response: The authors have revised this point in manuscript according to the Reviewer’s suggestion(L139).

103-27 Details on data analysis are missing. How many of the reads were of plant origin? How many reads were obtained in total? Was the sampling depth sufficient for analysis? Rarefied? A lot of basic information is missing in order to make a clear decision on the quality of the data provided.

Response: According to the reviewer's comments, the content has been supplemented in Supplementary material. Detailed information is attached to the table at the end of the document

139 Were both used and if so what were they exactly used for? Expand and explain.

Results

Response: The number of all sample sequences is leveled according to the minimum number of samples. After leveling, the average Good's coverage of each sample can still reach 99.09%. Taxonomic annotation of OTU species was performed by using RDP classifier(http://rdp.cme.msu.edu/, version 2.11) compared with Silva 16S rRNA gene database (v138). The confidence threshold was 70%. The community composition of each sample was calculated at different species classification levels. Bacteria were compared using the Silva 138 database;Fungi were compared using Unite 8.0.

143 According to the figures this is not the case. All have letter A above the columns.

Response: The author calculated the average and variance of tomatoes in different growth stages by SPSS software, and finally presented a significant table. The following tables are the significant differences between bacteria and fungi respectively. The authors have revised this point in manuscript according to the Reviewer’s suggestion(L173).

Treatment

Shannon

ACE

Seedling stage

3.99±0.12a

360.62±29.88b

Flower bud differentiation stage

4.44±0.49a

541.4±78.54a

Fruiting period

4.4±0.45a

560.85±7.19a

Table. Alpha diversity indexes of bacteria in root of tomatoes under different growth stages

Table. Alpha diversity indexes of fungi in root of tomatoes under different growth stages

Treatment

Shannon

ACE

Seedling stage

1.69±0.72ab

148.79±39.43b

Flower bud differentiation stage

0.9±0.01b

288.57±42.35a

Fruiting period

2.14±0.54a

357.44±82.61a

150 Unclear whether data set is weighed or unweighed.

Response: data set is unweighed.

Figure 1 Labelling very small so that reading is difficult even at 200%. Anosim or similar comparisons to find out if the separation is significant?

Response: The authors have revised this point in manuscript according to the Reviewer’s suggestion.(L173)

189 and throughout manuscript unclassified etc., this is not a phylum, please remove.

Response: The authors have revised this point in manuscript according to the Reviewer’s suggestion.(L190-305)

205-223 Genus names in italics.

Response: The authors have revised this point in manuscript according to the Reviewer’s suggestion.(L222-261)

205pp Unclassified is not a genus. Remove taxa that are not classified at taxon level. The whole section is largely a written form of figure 2 and therefore quite repetitive.

Response: The authors have revised this point in manuscript according to the Reviewer’s suggestion.(L236-261)

Discussion

297 hormones play

Response: The authors have revised this point in manuscript according to the Reviewer’s suggestion.(L299)

297-305 It would have been useful to measure some of these metabolomes/enzymes in parallel to the endophyte analysis.

Response: Thank you for your good comments.

338-43 The study has limited value as no plant metabolomes were measured and therefore, the presence of endophytes is purely discussed on the basis of other studies’ findings. I would have expected to have at least some key measures of the plant root to be included in the analysis.

At the moment, the study is quite one sided and limited. This is also evident from the short and limited discussion.

Response: Thank you for your good comments. Exactly, the authors descried  the data given in the report of the Illumina MiSeq. However, we found out what differences of endophytic microbial communities in tomatoes roots during different growing stages. Meanwhile, all the functions of the microbe were not our prediction, their functions were all verified and reported by the previous studied, the authors only summarized their functions based on the previous studies.

Conclusions

345-57 This is not a conclusion

Response: The authors have revised this point in manuscript according to the Reviewer’s suggestion(L347)

357-9 This is pure speculation at this point as the conditions of the plants were not measured other than the growth stage.

Response: There is literature showing that changes in microbial community composition can indicate changes in plant growth.

He, C.; Liu, C.; Liu, H.; Wang, W.; Hou, J.; Li, X. Dual Inoculation of Dark Septate Endophytes and Trichoderma Viride Drives Plant Performance and Rhizosphere Microbiome Adaptations of Astragalus Mongholicus to Drought. Environmental Microbiology, 2022, 24, 324–340. https://doi.org/10.1111/1462-2920.15878.

Cipriano, M. A. P.; Freitas-Iório, R. de P.; Dimitrov, M. R.; de Andrade, S. A. L.; Kuramae, E. E.; Silveira, A. P. D. da. Plant-Growth Endophytic Bacteria Improve Nutrient Use Efficiency and Modulate Foliar N-Metabolites in Sugarcane Seedling. Microorganisms, 2021, 9, 479. https://doi.org/10.3390/microorganisms9030479.

Rani, S.; Kumar, P.; Dahiya, P.; Maheshwari, R.; Dang, A. S.; Suneja, P. Endophytism: A Multidimensional Approach to Plant–Prokaryotic Microbe Interaction. Frontiers in Microbiology, 2022, 13. https://doi.org/10.3389/fmicb.2022.861235.

Data availability

371 Provide accession numbers for each growth stage

Response: The authors have revised this point in manuscript according to the Reviewer’s suggestion

Round 2

Reviewer 1 Report

Comments and Suggestions for Authors

The authors addressed well the comments of the reviewers, and the quality of the manuscript was improved; some minor aspects still need to be reviewed in the manuscript before accepting for publication.

Commentaries:

The new title proposed not really match with the main findings of the study, I suggest restructure the title.

This is an example: Endophytic microorganisms in tomato roots, changes in the structure and function of the community at different growing stages.

In lines 71-72, review the fragment is a bit repetitive “Different dominant microbes in tomatoes at various developmental stages have beneficial effects on their growth-promoting mechanisms in tomatoes [7].”; This is a propose: “Different dominant microbes present in tomatoes roots at various developmental stages have beneficial effects as growth-promoting [7].”

In line 91, “22.°C” must be “22 °C”

In line 108, “the cultivation medium” could be “the substrate”

In lines 111-114, complement with the objective of the following procedure “This was followed by a 1 min wash in 95% ethanol, a 14 min wash with agitation in sodium hypo-chlorite solution (1.4% active chlorine), a 10 s wash in 95% ethanol, followed by 10 rinses in sterile water with agitation”

In line 124, could be better “Tris-HCl”

Line 294, check if “Candidatus Solibacter” is correct

In lines 301-302, review “For example and Cyberlindnera were enriched in the GH stage” it could be “For example Cyberlindnera was enriched in the GH stage”

In line 377, add a space between “Leifsonia.were” and change the period by a comma.

In Conclusion, complement with information about how the findings of the study could be employed to improve tomato production or with perspectives of the study.

Author Response

Dear Editors and Reviewers:

Thank you for your letter and the reviewers’ comments concerning our manuscript entitled “Characteristics of endophytic microbial community structures in tomato roots under different growing stages” (Submission ID: microorganisms-2996270). Those comments are all valuable and very helpful for revising and improving our paper, as well as the important guiding significance tour researches. We have read comments carefully and have made correction which we hope meet with approval. Revised portion are marked in Green in the paper. The main corrections in the paper and the responds to the reviewer’s comments are as flowing:

Reviewers1

The authors addressed well the comments of the reviewers, and the quality of the manuscript was improved; some minor aspects still need to be reviewed in the manuscript before accepting for publication.

Commentaries:

The new title proposed not really match with the main findings of the study, I suggest restructure the title.

Response: The authors have revised this point in manuscript according to the Reviewer’s suggestion(L2)

This is an example: Endophytic microorganisms in tomato roots, changes in the structure and function of the community at different growing stages.

Response: The authors have revised this point in manuscript according to the Reviewer’s suggestion(L2)

In lines 71-72, review the fragment is a bit repetitive “Different dominant microbes in tomatoes at various developmental stages have beneficial effects on their growth-promoting mechanisms in tomatoes [7].”; This is a propose: “Different dominant microbes present in tomatoes roots at various developmental stages have beneficial effects as growth-promoting [7].”

Response: The authors have revised this point in manuscript according to the Reviewer’s suggestion(L68)

In line 91, “22.°C” must be “22 °C”

Response: The authors have revised this point in manuscript according to the Reviewer’s suggestion(L86)

In line 108, “the cultivation medium” could be “the substrate”

Response: The authors have revised this point in manuscript according to the Reviewer’s suggestion(L114)

In lines 111-114, complement with the objective of the following procedure “This was followed by a 1 min wash in 95% ethanol, a 14 min wash with agitation in sodium hypo-chlorite solution (1.4% active chlorine), a 10 s wash in 95% ethanol, followed by 10 rinses in sterile water with agitation”

 Response: The authors have revised this point in manuscript according to the Reviewer’s suggestion(L117)

In line 124, could be better “Tris-HCl”

 Response: The authors have revised this point in manuscript according to the Reviewer’s suggestion(L129)

Line 294, check if “Candidatus Solibacter” is correct

Response: Thank you for your good comments. NCBI BLAST name: bacteria, Rank: genus.

Ward, N. L.; Challacombe, J. F.; Janssen, P. H.; Henrissat, B.; Coutinho, P. M.; Wu, M.; Xie, G.; Haft, D. H.; Sait, M.; Badger, J.; Barabote, R. D.; Bradley, B.; Brettin, T. S.; Brinkac, L. M.; Bruce, D.; Creasy, T.; Daugherty, S. C.; Davidsen, T. M.; DeBoy, R. T.; Detter, J. C.; Dodson, R. J.; Durkin, A. S.; Ganapathy, A.; Gwinn-Giglio, M.; Han, C. S.; Khouri, H.; Kiss, H.; Kothari, S. P.; Madupu, R.; Nelson, K. E.; Nelson, W. C.; Paulsen, I.; Penn, K.; Ren, Q.; Rosovitz, M. J.; Selengut, J. D.; Shrivastava, S.; Sullivan, S. A.; Tapia, R.; Thompson, L. S.; Watkins, K. L.; Yang, Q.; Yu, C.; Zafar, N.; Zhou, L.; Kuske, C. R. Three Genomes from the PhylumAcidobacteriaProvide Insight into the Lifestyles of These Microorganisms in Soils. Applied and Environmental Microbiology, 2009, 75, 2046–2056. https://doi.org/10.1128/aem.02294-08.

In lines 301-302, review “For example and Cyberlindnera were enriched in the GH stage” it could be “For example Cyberlindnera was enriched in the GH stage”

Response: The authors have revised this point in manuscript according to the Reviewer’s suggestion(L309)

In line 377, add a space between “Leifsonia.were” and change the period by a comma.

Response: The authors have revised this point in manuscript according to the Reviewer’s suggestion(L382)

In Conclusion, complement with information about how the findings of the study could be employed to improve tomato production or with perspectives of the study.

Reviewer 3 Report

Comments and Suggestions for Authors

The authors have conducted a partial revision of the manuscript. While some aspects were revised, other aspects were stated as revised but upon inspection were not, which is somewhat frustrating.

Main comments:

The authors are still not sufficiently recognizing that their data are based on relative abundances and not absolute abundances. Therefore, results may be overinterpreted. Since the authors have not offered to back up their data with absolute quantification methods, they have to more clearly state the limitations of their study in the discussion.

Additional information has been provided for the sequence analysis but this section is still incomplete and needs to be fully addressed (see detailed comments below).

Specific comments:

Title: Do you mean community structures? Functions are predicted so 

"Endophytic microorganisms can assemble in various community structures  with diverse predicted functions in tomato roots at different growing stages"

Abstract:

L18 Relative abundance

L22 revise sentence, prediction predicted

Introduction:

L53 Still not corrected.

L69 Still not addressed

Materials and Methods:

L94 use past tense

L97 Not addressed

L108 Not addressed

L115 Beckers reference missing

L135-6 Incomplete sentence

L137-8 Unclear, removed singletons?

L139 Unclear

L135-43 For steps 1-3, what software or tools were used?

L144 what tools used?

L145 Unclear, rarefied? If so, what tool used?

L150 Unclear, no species level identification possible.

L164-6 Specify what steps were analysed with these online tools.

Revision incomplete: How many of the reads were of plant origin? How many reads were obtained in total? Was the sampling depth sufficient for analysis? A lot of basic information is still missing in order to make a clear decision on the quality of the data provided.

Results:

Figure 1 Again, revision requests were ignored. What algorithm was used? Weighted or unweighted? Unifrac, Jaccard, Bray?

Discussion:

All results are based on relative abundances, what is the consequence of the findings and how could this potentially weaken the findings?

Functional assignments are based on predictions only. Bacterial genus identification doe snot mean that the found genera actually have the predicted function. This appears to be a critical overinterpretation of the data. This needs to be addressed before the manuscript can be considered for publication.

Comments on the Quality of English Language

Only a few minor sections that need improvements (see specific comments).

Author Response

Dear Editors and Reviewers:

Thank you for your letter and the reviewers’ comments concerning our manuscript entitled “Characteristics of endophytic microbial community structures in tomato roots under different growing stages” (Submission ID: microorganisms-2996270). Those comments are all valuable and very helpful for revising and improving our paper, as well as the important guiding significance tour researches. We have read comments carefully and have made correction which we hope meet with approval. Revised portion are marked in Green in the paper. The main corrections in the paper and the responds to the reviewer’s comments are as flowing:

Reviewers3

The authors have conducted a partial revision of the manuscript. While some aspects were revised, other aspects were stated as revised but upon inspection were not, which is somewhat frustrating.

Main comments:

The authors are still not sufficiently recognizing that their data are based on relative abundances and not absolute abundances. Therefore, results may be overinterpreted. Since the authors have not offered to back up their data with absolute quantification methods, they have to more clearly state the limitations of their study in the discussion.

Additional information has been provided for the sequence analysis but this section is still incomplete and needs to be fully addressed (see detailed comments below).

Specific comments:

Title: Do you mean community structures? Functions are predicted so 

Response: The authors have revised this point in manuscript according to the Reviewer’s suggestion(L2)

"Endophytic microorganisms can assemble in various community structures  with diverse predicted functions in tomato roots at different growing stages"

Abstract:

L18 Relative abundance

Response: The authors have revised this point in manuscript according to the Reviewer’s suggestion(L15)

L22 revise sentence, prediction predicted

Response: The authors have revised this point in manuscript according to the Reviewer’s suggestion(L19)

Introduction:

L53 Still not corrected.

Response: The authors have revised this point in manuscript according to the Reviewer’s suggestion(L48)

L69 Still not addressed

Response: The authors have revised this point in manuscript according to the Reviewer’s suggestion(L65-69)

Kuroha, T.; Tokunaga, H.; Kojima, M.;et al. Functional Analyses of LONELY GUY Cytokinin-Activating Enzymes Reveal the Importance of the Direct Activation Pathway in Arabidopsis. The Plant Cell, 2009, 21, 3152–3169. https://doi.org/10.1105/tpc.109.068676.

Materials and Methods:

L94 use past tense

Response: The authors have revised this point in manuscript according to the Reviewer’s suggestion(L83-90)

L97 Not addressed

Response: The authors have revised this point in manuscript according to the Reviewer’s suggestion(L97-109)

L108 Not addressed

Response: The authors have revised this point in manuscript according to the Reviewer’s suggestion(L97-109)

L115 Beckers reference missing

Response: The authors have revised this point in manuscript according to the Reviewer’s suggestion(L484)

Add

L135-6 Incomplete sentence

Response: The authors have revised this point in manuscript according to the Reviewer’s suggestion(L117-121)

L137-8 Unclear, removed singletons?

Response: The authors have revised this point in manuscript according to the Reviewer’s suggestion. Remove singletons, which are sequences that do not have duplicates. The specific operation page used here is (http://drive5.com/usearch/manual/singletons.html)

L139 Unclear

Response: The authors have revised this point in manuscript according to the Reviewer’s suggestion(144-162). The fourth step is to map all optimized sequences to OTU representative sequences. Each sequence is compared with the representative sequence of the OTU. Sequences showing more than 97% similarity with any representative sequence are selected for generating the OTU table. Our software uses Uparse (version 7.0.1090 http://drive5.com/uparse/)

L135-43 For steps 1-3, what software or tools were used?

Response: The authors have revised this point in manuscript according to the Reviewer’s suggestion(L144-162). We have marked the corresponding website platform used in the OTU clustering step after the corresponding step. If there is anything unclear or if you have any questions, please feel free to let me know. I would be happy to provide further explanations.

L144 what tools used?

Response: The authors have revised this point in manuscript according to the Reviewer’s suggestion(L129). we use RDP classifier version 2.11(http://RDP), set the confidence threshold to 0.7, and classify it against the 16s rRNA gene database Silva v138 (http://www.arb-Silva.de).

L145 Unclear, rarefied? If so, what tool used?

Response: The authors have revised this point in manuscript according to the Reviewer’s suggestion(L129). we use RDP classifier version 2.11(http://RDP), set the confidence threshold to 0.7, and classify it against the 16s rRNA gene database Silva v138 (http://www.arb-Silva.de).

L150 Unclear, no species level identification possible.

Response: The authors have revised this point in manuscript according to the Reviewer’s suggestion(L129). The RDP classifier we choose here is a tool based on Bayesian algorithm to identify the 16SrRNA sequences of bacteria and archaea. It compares unknown sequences with known taxa to provide phylogenetic classification. Use the Silvav138 database for comparison. The output will provide the possible classification of each sequence and the confidence score corresponding to each classification according to the selected database and confidence threshold.

L164-6 Specify what steps were analysed with these online tools.

Response: The authors have revised this point in manuscript according to the Reviewer’s suggestion(L146-162).

Revision incomplete: How many of the reads were of plant origin? How many reads were obtained in total? Was the sampling depth sufficient for analysis? A lot of basic information is still missing in order to make a clear decision on the quality of the data provided.

Response: The authors have revised this point in manuscript according to the Reviewer’s suggestion. Endophytic bacteria: Diversity data analysis of 9 samples was completed, and a total of 209059 optimized sequences, 78610981 bases, and an average sequence length of 376 bp were obtained. Endophytic fungi: Completed diversity data analysis of 9 samples, and obtained a total of 1,234,928 optimized sequences, 287,046,596 bases, and an average sequence length of 232 bp.

Endophytic bacteria:

Amplified Region

Insert size(bp)

Sequencing Length

Raw Reads

Total Base(bp)

799F_1193R

394

PE300

209059*2

125853518

Amplified Region

Samples

Sequences

Bases(bp)

Average Length

799F_1193R

9

209059

78610981

376

Endophytic fungi:

Amplified Region

Insert size(bp)

Sequencing Length

Raw Reads

Total Base(bp)

ITS1F_ITS2R

300

PE300

1234928*2

743426656

Amplified Region

Samples

Sequences

Bases(bp)

Average Length

ITS1F_ITS2R

9

1234928

287046596

232

Results:

Figure 1 Again, revision requests were ignored. What algorithm was used? Weighted or unweighted? Unifrac, Jaccard, Bray?

Response: The authors have revised this point in manuscript according to the Reviewer’s suggestion(L206). Data set is unweighed. The distance algorithm used is Bray, and the difference test between groups is Adonis.

Discussion:

All results are based on relative abundances, what is the consequence of the findings and how could this potentially weaken the findings?

Response: In this study, we calculated the relative abundance ratios within the same sample and compared them to identify the dominant and endemic flora. By analyzing the changes in these ratios across different groups, we can observe how the flora varies across various growth periods. This analysis is particularly useful for understanding the roles that microorganisms play at different stages of development.However, there are some limitations to consider. Since we are not comparing absolute quantities, an apparent increase in abundance could actually correspond to a decrease in quantity. Nonetheless, the focus of this paper is on the shifts in community composition. We specifically examine the different growth stages of tomato roots, where not only endemic bacteria and fungi are present, but also the functions of enriched microorganisms, which primarily enhance plant growth, stress resistance, and nutrient cycling. Our results also demonstrate that endophytes in tomato roots perform different functions at various growth stages. Therefore, the use of relative abundance ratios in our analysis does not compromise our findings.

Functional assignments are based on predictions only. Bacterial genus identification doe snot mean that the found genera actually have the predicted function. This appears to be a critical overinterpretation of the data. This needs to be addressed before the manuscript can be considered for publication.

Response: Exactly, the authors descried  the data given in the report of the Illumina MiSeq. However, we found out what differences of endophytic microbial communities in tomatoes roots during different growing stages. Meanwhile, all the functions of the microbe were not our prediction, their functions were all verified and reported by the previous studied, the authors only summarized their functions based on the previous studies.
